# Health impact of monoclonal gammopathy of undetermined significance (MGUS) and monoclonal B-cell lymphocytosis (MBL): findings from a UK population-based cohort

Maxine JE Lamb [1] ,[1] Alexandra Smith [1] ,[1] Daniel Painter [1] ,[1] Eleanor Kane [1] ,[1] Timothy Bagguley [1] ,[1] Robert Newton [1] ,[1,2] Debra Howell [1] ,[1] Gordon Cook,[3] Ruth de Tute,[4] Andrew Rawstron,[4] Russell Patmore,[5] Eve Roman [1] [1]

For numbered affiliations see end of article.

**Correspondence to**
Professor Eve Roman;
eve.roman@york.ac.uk

## ABSTRACT

**Objective** To examine mortality and morbidity patterns before and after premalignancy diagnosis in individuals with monoclonal gammopathy of undetermined significance (MGUS) and monoclonal B-cell lymphocytosis (MBL) and compare their secondary healthcare activity to that of the general population.

**Design** Population-based patient cohort, within which each patient is matched at diagnosis to 10 age-matched and sex-matched individuals from the general population. Both cohorts are linked to nationwide information on deaths, cancer registrations and Hospital Episode Statistics.

**Setting** The UK's Haematological Malignancy Research Network, which has a catchment population of around 4 million served by 14 hospitals and a central diagnostic laboratory.

**Participants** All patients newly diagnosed during 2009–2015 with MGUS (n=2193) or MBL (n=561) and their age and sex-matched comparators (n=27 538).

**Main outcome measures** Mortality and hospital inpatient and outpatient activity in the 5 years before and 3 years after diagnosis.

**Results** Individuals with MGUS experienced excess morbidity in the 5 years before diagnosis and excess mortality and morbidity in the 3 years after diagnosis. Increased rate ratios (RRs) were evident for nearly all clinical specialties, the largest, both before and after diagnosis, being for nephrology (before RR=4.29, 95% CI 3.90 to 4.71; after RR=13.8, 95% CI 12.8 to 15.0) and rheumatology (before RR=3.40, 95% CI 3.18 to 3.63; after RR=5.44, 95% CI 5.08 to 5.83). Strong effects were also evident for endocrinology, neurology, dermatology and respiratory medicine. Conversely, only marginal increases in mortality and morbidity were evident for MBL.

**Conclusions** MGUS and MBL are generally considered to be relatively benign, since most individuals with monoclonal immunoglobulins never develop a B-cell malignancy or any other monoclonal protein-related organ/tissue-related disorder. Nonetheless, our findings offer strong support for the view that in some individuals, monoclonal gammopathy has the potential to cause

### Strengths and limitations of this study

⇒ Data are from an established population-based cohort within which all haematological malignances and related clonal disorders are diagnosed, monitored and coded at a single laboratory.
⇒ Providing nationally generalisable data, all diagnoses are included, and complete follow-up is achieved via linkage to nationwide administrative datasets.
⇒ The age-matched and sex-matched general population cohort enables baseline activity and rate ratios to be calculated, both before and after premalignancy detection.
⇒ Analyses are constrained by the fact that hospital episodes statistics are primarily collected for administrative and clinical purposes and not for research.

systemic disease resulting in wide-ranging organ/tissue damage and excess mortality.

## BACKGROUND

Monoclonal gammopathy of undetermined significance (MGUS) and monoclonal B-cell lymphocytosis (MBL) are premalignant monoclonal B-cell disorders, the former progressing to myeloma at a rate of around 1% per year[1 2] and the latter to chronic lymphocytic leukaemia (CLL) at around 2% per year.[3 4] Diagnosed more frequently in men than women and people over 60 years of age,[5 6] overt symptoms of haematological malignancy are, by definition, absent in both MBL and MGUS.[7] Accordingly, although some premalignant disorders are found coincidentally during routine health checks, others are identified during diagnostic work-up investigations; MGUS during the course of tests applied to detect a range

of potential conditions and illnesses[8 9] and MBL during episodes of unexplained lymphocytosis.[4 10]

In addition to the association with haematological malignancy, individuals with MGUS or MBL sometimes experience higher than expected levels of mortality and morbidity that are independent of cancer.[4 8 11–16] Indeed, although most individuals with these disorders suffer no obvious ill effects, interest in their relationship with other comorbidities has increased markedly in recent years, MBL largely in relation to its potential to impact on the immune response[17] and MGUS due to the systemic organ and tissue damage that can be caused by monoclonal immunoglobulins secreted by the abnormal B-cell clone.[18] Hitherto, however, most information about these associations has been derived either from case–control studies established to look at risk factors for disease development (eg, family history of disease) and additional tests applied to specific patient groups (eg, patients with kidney disease) or cohort studies that track individuals with either MGUS or MBL forwards in time from their diagnosis.[5 13 18] However, despite the undoubted interest in the sequence of health events, as far as we are aware, no systematic population-based investigations of the comorbidity patterns that precede and succeed a diagnosis of either MGUS or MBL have been undertaken in the same cohort.

With a view to shedding light on the health events occurring before and after the diagnosis of MGUS and MBL, the present report uses data from an established UK population-based patient cohort of haematological malignancies and related disorders to examine the comorbidity patterns of individuals with these premalignancy clonal disorders (MGUS=2193, MBL=561). To enable effect size quantification, these patterns are compared with the baseline activity of an individually age-matched and sex-matched (10 per patient) general population comparison cohort.

## METHODS

Cases are from the Haematological Malignancy Research Network (HMRN; www.HMRN.org), a specialist UK registry established in 2004 to provide robust generalisable data to inform contemporary clinical practice and research across the country as a whole.[19 20] HMRN operates under a legal basis that permits data to be collected directly from health records without explicit consent. Set within a catchment population of around four million that is served by 14 hospitals and has a socioeconomic profile which is broadly representative of the UK as a whole, all haematological cancers and related conditions are diagnosed and coded by clinical specialists at a single integrated haematopathology laboratory, the Haematological Malignancy Diagnostic Service (www.HMDS. info), using standardised diagnostic criteria and the latest WHO International Statistical Classification of Diseases third revision (ICD-O-3) classification.[7] Specifically, in relation to the present report, which covers diagnoses

made during 2009–2015, MBL was defined by a peripheral blood monoclonal B-cell count $<5\times10^9$/L in individuals with no other features of a B-cell lymphoproliferative disorder, in MGUS by a serum paraprotein less than 30 g/L and in those where a bone marrow examination was considered necessary following clinical examination, a clonal bone marrow plasma cells/lymphoplasmacytic infiltration of less than 10%. Hence, within the HMRN region, as in other diagnostic settings,[21 22] invasive bone marrow examinations in patients with MGUS are generally only carried when laboratory or clinical features are indicative of an underlying plasma cell neoplasm, lymphoproliferative disorder, monoclonal immunoglobulin deposition disease (eg, amyloidosis) or conditions like polyneuropathy, organomegaly, endocrinopathy, monoclonal gammopathy and skin changes (POEMS) syndrome.[7]

To facilitate comparisons with unaffected individuals, HMRN also has a general population cohort. To create this 'control' cohort, all patients diagnosed during 2009–2015 with a haematological malignancy or related clonal disorder (n=18 127) were individually matched on sex and age at the point of diagnosis to 10 randomly selected individuals from the same catchment population. All controls were assigned a serial number that linked them to their matched case and a 'pseudodiagnosis' date that corresponded to their matched case's diagnosis date. Individuals in the patient cohort and the comparison cohort are linked to the same nationwide information on deaths, cancer registrations and Hospital Episode Statistics (HES). At the point of selection and matching, all controls were resident in the HMRN region and none had a previous cancer registration for a haematological malignancy.[23 24] Hence, for the control cohort (in contrast to the patient cohort) no additional health information outside that contained within national administrative datasets was available.

Using similar methods to those previously described,[23–25] associations with hospital inpatient activity (HES admitted patient care) and outpatient activity (HES outpatient (HES-OP)) in the 5 years prior to diagnosis/pseudodiagnosis through to the 3 years after diagnosis were investigated. HES inpatient data contain ICD-10 codes derived from discharge summaries[26] and associations with these were examined in relation to the 17 specific conditions in the Charlson Comorbidity Index.[27–29] By contrast, HES-OP data contain details about the type of outpatient attendance, the majority being linked to consultant specialty codes (eg, ophthalmology, rheumatology), with the remainder largely comprising routine follow-up/monitoring, nurse-led clinic attendances (eg, anticoagulant clinics) and consultations with allied health professionals (eg, podiatry).

This report includes all patients (cases) who were newly diagnosed with either MGUS (n=2193) or MBL (n=561) between 1 January 2009 and 31 December 2015 and their matched controls (n=27 538); individuals diagnosed with a haematological cancer within 6 months of their MGUS/

**Table 1** Characteristics of patients diagnosed with monoclonal gammopathy of undetermined significance (MGUS) or monoclonal B-cell lymphocytosis (MBL) and their corresponding controls: HMRN diagnoses 2009–2015

| | MGUS | | MBL | |
| --- | --- | --- | --- | --- |
| | Cases Number (%) | Controls Number (%) | Cases Number (%) | Controls Number (%) |
| Total | 2193 (100) | 21 928 (100) | 561 (100) | 5610 (100) |
| Gender | | | | |
| Male | 1131 (51.6) | 11 308 (51.6) | 328 (58.5) | 3280 (58.5) |
| Female | 1062 (48.4) | 10 620 (48.4) | 233 (41.5) | 2330 (41.5) |
| Age (years) | | | | |
| <60 | 347 (15.8) | 3470 (15.8) | 85 (15.2) | 850 (15.2) |
| 60–70 | 500 (22.8) | 5000 (22.8) | 153 (27.3) | 1530 (27.3) |
| 70–80 | 782 (35.7) | 7820 (35.7) | 194 (34.6) | 1940 (34.6) |
| ≥80 | 564 (25.7) | 5638 (25.7) | 129 (23.0) | 1290 (23.0) |
| Median (IQR) | 73.4 (64.6–80.2) | 73.4 (64.6–80.2) | 72.1 (64.5–79.2) | 72.1 (64.5–79.2) |
| Overall survival* | | | | |
| 5 years (95% CI) | 71.9 (69.5 to 74.1) | 80.1 (79.4 to 80.7) | 77.2 (72.5 to 81.2) | 80.7 (79.4 to 81.9) |
| Relative survival | | | | |
| 5 years (95% CI) | 90.2 (87.4 to 92.4) | 99.5 (98.7 to 99.8) | 94.3 (86.3 to 97.7) | 98.5 (96.8 to 99.3) |
| Progressions and transformations* | | | | |
| Total | 75 (3.4) | | 140 (25.0) | |
| Myeloma | 48 (2.2) | – | – | – |
| CLL† | 1 (0.05) | – | 137 (24.4) | – |
| Other | 26 (1.2) | – | 3 (0.5) | – |
| Non-haematological cancer registrations* | | | | |
| Total | 185 (8.4) | 1748 (8.0) | 58 (10.3) | 494 (8.8) |
| Lung | 31 (1.5) | 190 (0.9) | 4 (0.7) | 59 (1.1) |
| Prostate‡ | 20 (1.8) | 178 (1.6) | 3 (0.9) | 56 (1.7) |
| Colorectal | 19 (0.9) | 166 (0.8) | 6 (1.1) | 40 (0.7) |
| Breast‡ | 15 (1.4) | 135 (1.3) | 4 (1.7) | 16 (0.7) |
| Comorbidity score before MGUS/MBL diagnosis§ | | | | |
| 0 | 1256 (57.3) | 15 893 (72.5) | 419 (74.7) | 4115 (73.4) |
| 1 | 506 (23.1) | 3617 (16.5) | 84 (15.0) | 898 (16.0) |
| ≥2 | 431 (19.7) | 2418 (11.0) | 58 (10.3) | 597 (10.6) |
| | p<0.001 | | p=0.78 | |

*Followed up until 15 March 2017.
†Chronic lymphocytic leukaemia (CLL).
‡Prostate for men only, breast for women only.
§≥1 hospital admission with a diagnosis of 1 of the 17 conditions included in the Charlson Comorbidity Index[25–27] during the 5 years to 1 month before MGUS/MBL diagnosis (patients)/pseudodiagnosis (controls).
HMRN, Haematological Malignancy Research Network.

MBL diagnosis were considered ineligible. All cases and controls were followed up for cancer registration and death until March 2017 and hospital activity (inpatient and outpatient) until March 2016. Additionally, progressions and/or transformations among cases were identified through HMDS up to March 2017. Data were summarised using standard methods. Overall survival, hospital activity and rate ratios (RRs) were calculated using time-to-event analyses. The Stata program 'strel' was used to estimate relative survival (RS), using age-specific and sex-specific background mortality rates from national life tables.[30 31] All analyses were conducted using Stata V.16.0.

### Patient and public involvement (PPI)

PPI is integral to HMRN and takes place via a dedicated patient partnership, overseen by a lay committee. Patients from the partnership are involved in identifying key research questions and participate in all our funding applications. Furthermore, patients and their relatives routinely take part in the dissemination of HMRN's

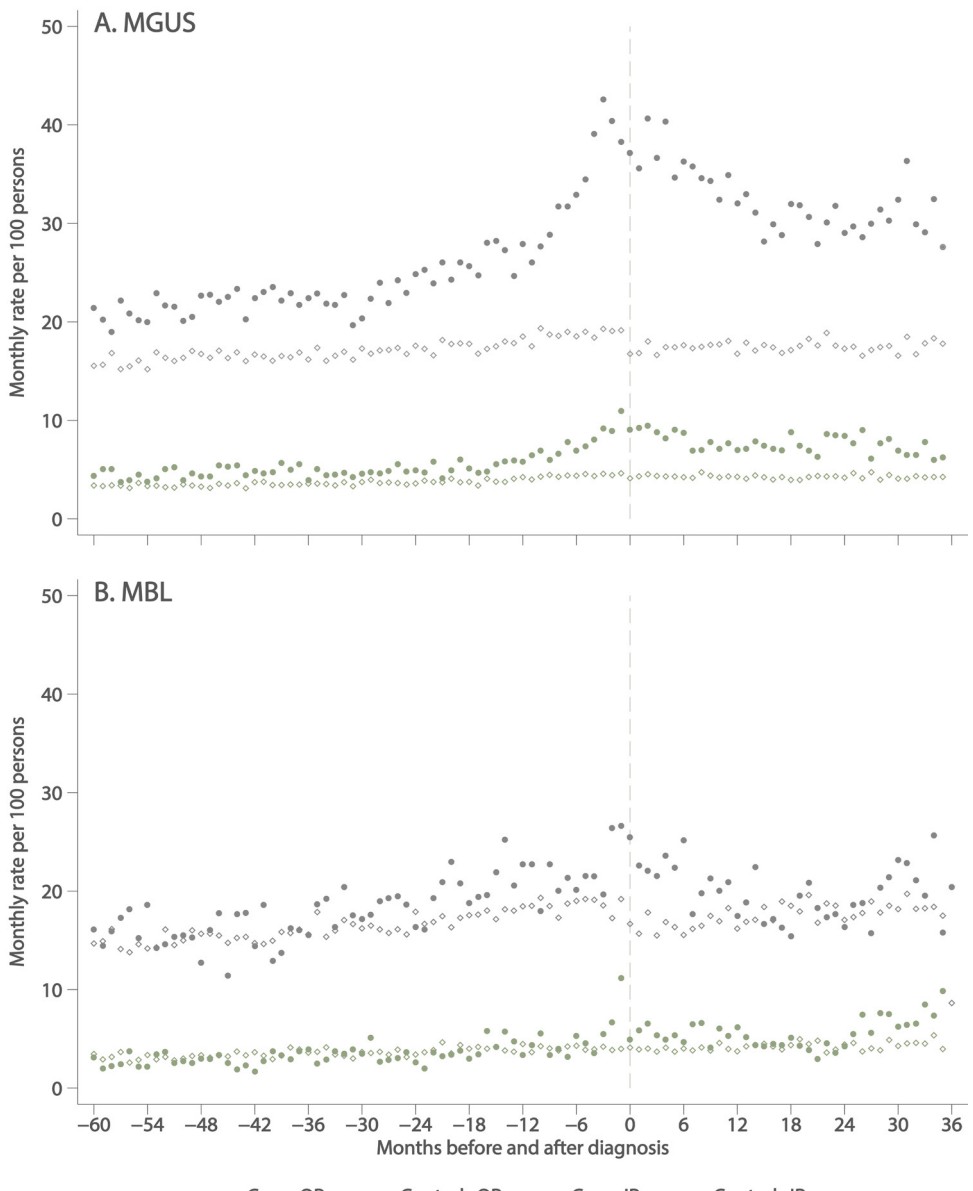

**Figure 1** Inpatient (IP) and outpatient (OP) monthly hospital visit activity rates 5 years before to 3 years after diagnosis (cases)/pseudodiagnosis (controls) excluding haematology attendances. (A) Monoclonal gammopathy of undetermined significance (MGUS) and (B) monoclonal B-cell lymphocytosis (MBL) diagnosed during 2009–2015.

findings, which also occurs via our project website: www.yhhn.org.

## RESULTS

Characteristics of individuals with either MGUS or MBL are summarised alongside their corresponding controls in table 1. Both MGUS and MBL were more commonly diagnosed in males (51.6% MGUS, 58.5% MBL) and individuals over the age of 70 years (median diagnostic age 73.4 years for MGUS and 72.1 years for MBL). At 90.2% (95% CI 87.4% to 92.4%), the 5-year RS of patients diagnosed with MGUS was significantly lower (p<0.01) than that of their general population controls (RS 99.5%, 95% CI 98.7% to 99.8%). For MBL, the difference was far less marked, the 5-year RS being 94.3% (95% CI 86.3%

to 97.7%) for MBL cases and 98.5% (95% CI 96.8% to 99.3%) for their controls. Interestingly, the observed variations in blood cancer progression were in the opposite direction to those seen for mortality: 75 (3.4%) patients with MGUS were diagnosed with a haematological malignancy (48/75 were myelomas) before April 2017, compared with 140 (25.0%) patients with MBL (137/140 were CLLs). No significant differences were, however, evident for non-haematological cancer registrations, although the frequencies were slightly higher for MGUS (8.4%) and MBL (10.3%) than for their corresponding controls (8.0% and 8.8%, respectively).

With respect to comorbidity, in the years leading up to diagnosis, patients with MGUS were significantly more likely than their corresponding controls to have

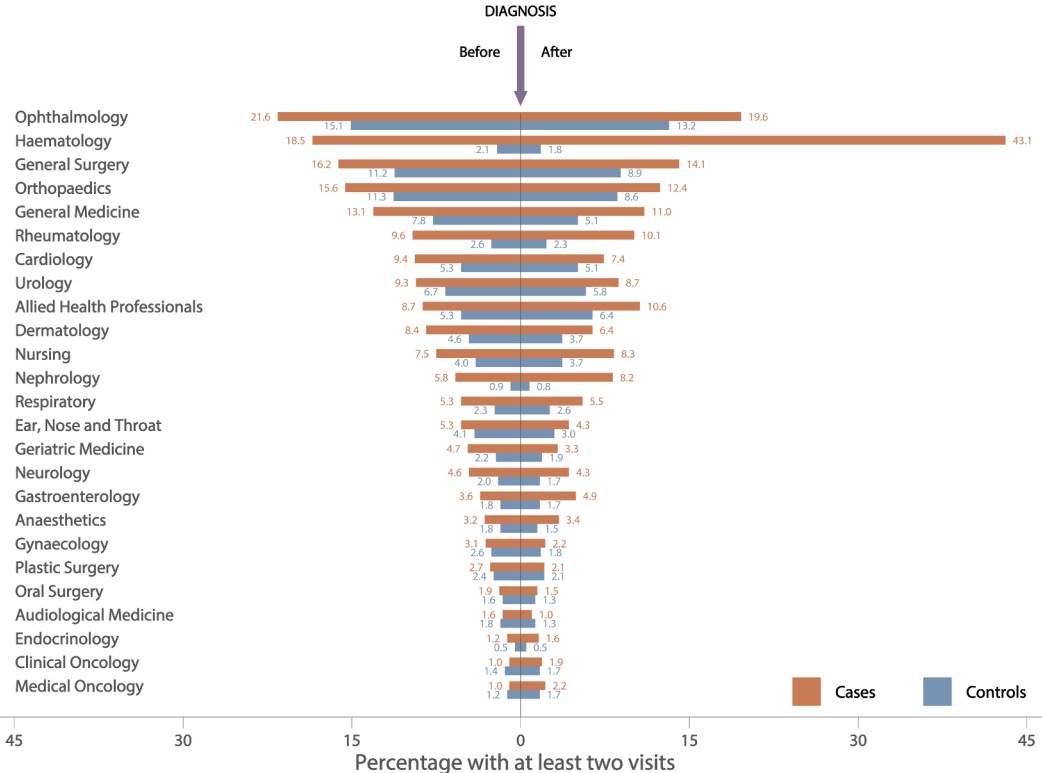

**Figure 2** Percentage of cases and controls with at least two specialty-specific outpatient visits in the 3 years before and after diagnosis of monoclonal gammopathy of undetermined significance. Top 25 recorded specialties, with visits within 1 month of diagnosis/pseudodiagnosis excluded.

a record of at least 1 of the 17 comorbidities specified in the Charlson Comorbidity Index[27–29] recorded in their discharge summaries, but no differences between MBL cases and their controls were evident (table 1). More information about hospital activity patterns of cases with MGUS/MBL and their general population controls is shown in figure 1, which shows inpatient and outpatient activity (excluding haematology) during the 5 years before and the 3 years after diagnosis of MGUS (figure 1A) and MBL (figure 1B). In the period before diagnosis, patients with MGUS (figure 1A) had consistently higher outpatient activity rates than their controls, the disparity increasing markedly during the 18 months leading up to the formal diagnosis of MGUS by haematopathology where it remained high for about 12 months, before gradually falling and levelling out at a higher level than before diagnosis. Although less pronounced, a similar pattern is evident in inpatient data. With smaller numbers and more scatter, variations in outpatient and inpatient activity in MBL are less evident (figure 1B).

Figure 2 shows outpatient attendance frequencies (at least two specialty-specific visits) in the 3 years before and in the 3 years after MGUS diagnosis for the top 25 clinical specialties, visits within 1 month (±) of diagnosis/pseudodiagnosis are excluded. As is evident from the plot, the increased outpatient activity seen among cases (figure 1) occurs across a range of clinical specialties, the highest frequencies occurring in ophthalmology, haematology, general surgery, orthopaedics, general (internal)

medicine and rheumatology. However, excluding haematology where, as expected, attendances increased markedly just before and after MGUS diagnosis, the largest RRs both before (figure 3A) and after (figure 3B) diagnosis were for nephrology (before diagnosis RR=4.29, 95% CI 3.90 to 4.71; after diagnosis RR=13.8, 95% CI 12.8 to 15.0) and rheumatology (before diagnosis RR=3.40, 95% CI 3.18 to 3.63; after diagnosis RR=5.44, 95% CI 5.08 to 5.83). Other significant associations (p<0.05) with RR point estimates above 2.0 were evident for endocrinology, neurology and respiratory medicine, as well as for the nurse-led monitoring activities which form part of ongoing clinical care across a range of specialties.

MGUS data are stratified by subtype in table 2. Accounting for around two-thirds of the total (n=1471; 67.0%), the IgG subtype dominates, followed by IgM (n=350; 16.0%) and IgA (n=266; 12.1%). The remaining 106 (4.8%) 'other' category comprise a mix of subtypes: light chain only (n=60), IgG+IgM (n=17), IgG+IgA (n=6), IgA+IgM (n=1), IgE (n=2) and not recorded (n=20). As expected, progression to myeloma in the 3 years following MGUS diagnosis was largely restricted to the IgG and IgA subtypes. The age distributions, 5-year survival estimates (overall and relative) and non-haematological malignancy frequencies of the main subtypes were broadly similar, although patients in the combined 'other' category tended to be slightly older and to fare less well (RS=77.6%, 95% CI 62.6% to 87.2%). The numbers of patients in the individual groups were, however, too

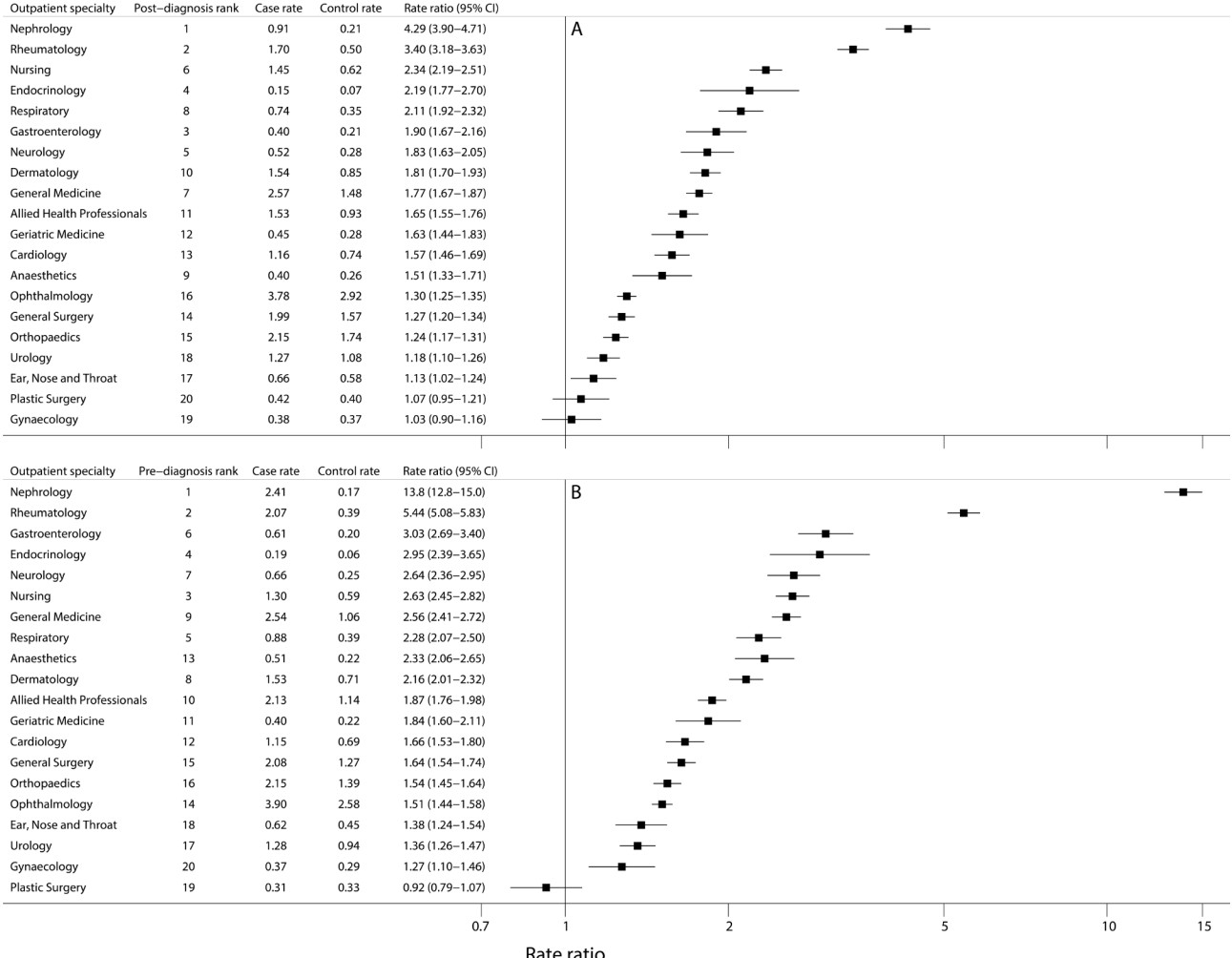

**Figure 3** Monthly outpatient attendance rates (per 100 persons) in cases and controls and rate ratios by outpatient specialty with at least two visits (A) in the 3 years before diagnosis and (B) in the 3 years after diagnosis of monoclonal gammopathy of undetermined significance.

sparse to examine the data in greater depth. Finally, during the study period (2009–2015), in addition to the detection of a paraprotein in peripheral blood, around 70% (1527/2193) of patients with MGUS in our cohort had a confirmatory bone marrow examination taken to exclude an underlying neoplasm. The patient characteristics and secondary care activity patterns of those who had bone marrow examinations were, however, broadly similar to those who did not (data not shown).

## DISCUSSION

Including data on nearly 3000 cases with premalignant clonal disorders and 10 times as many age-matched and sex-matched general population controls, this large UK record-linkage study found that individuals with MGUS not only experienced excess mortality and morbidity after diagnosis, but also excess morbidity in the 5 years before diagnosis. By contrast, only marginal increases in mortality and morbidity were evident for MBL, none of which were consistent or varied significantly from the general population. Interestingly, progression patterns were in the

opposite direction: in the years following detection of a premalignant clonal disorder, 3.4% (n=75/2193) of those with MGUS developed a haematological malignancy (48 of which were myelomas) before April 2017, compared with 25.0% (n=140/561) of those with MBL (137 of which were CLLs).

The elevated mortality and morbidity following a diagnosis of MGUS, which seems largely independent of progression to cancer, is consistent with reports relating to the potential clinical significance of this disorder.[11–18] Corresponding to the period of diagnostic work-up, our data also demonstrate the pronounced increase in hospital activity in the months surrounding MGUS diagnosis, the highest activity being observed in the 6 months before and 6 months after diagnosis. Of more importance, perhaps, the analyses clearly show that hospital activity in people subsequently diagnosed with MGUS is often elevated many years before diagnosis: excesses being observed in specialties covering most organ and tissue systems including nephrology, endocrinology, neurology, rheumatology, gastroenterology, dermatology

**Table 2** Characteristics of patients diagnosed with monoclonal gammopathy of undetermined significance (MGUS) and their corresponding controls, stratified by MGUS subtype:HMRN diagnoses 2009–2015

| | MGUS subtype | | | | | | | |
| | IgG | | IgM | | IgA | | Other* | |
| | Cases Number (%) | Controls Number (%) | Cases Number (%) | Controls Number (%) | Cases Number (%) | Controls Number (%) | Cases Number (%) | Controls Number (%) |
|---|---|---|---|---|---|---|---|---|
| Total | 1471 (100.0) | 14 709 (100.0) | 350 (100.0) | 3499 (100.0) | 266 (100.0) | 2660 (100.0) | 106 (100.0) | 1060 (100.0) |
| **Gender** | | | | | | | | |
| Male | 745 (50.6) | 7449 (50.6) | 177 (50.6) | 1769 (50.6) | 145 (54.5) | 1450 (54.5) | 64 (60.4) | 640 (60.4) |
| Female | 726 (49.4) | 7260 (49.4) | 173 (49.4) | 1730 (49.4) | 121 (45.5) | 1210 (45.5) | 42 (39.6) | 420 (39.6) |
| **Age (years)** | | | | | | | | |
| <60 | 239 (16.2) | 2390 (16.2) | 50 (14.3) | 500 (14.3) | 48 (18.0) | 480 (18.0) | 10 (9.4) | 100 (9.4) |
| 60–70 | 336 (22.8) | 3360 (22.8) | 80 (22.9) | 800 (22.9) | 56 (21.1) | 560 (21.1) | 28 (26.4) | 280 (26.4) |
| 70–80 | 514 (34.9) | 5140 (34.9) | 130 (37.1) | 1300 (37.2) | 104 (39.1) | 1040 (39.1) | 34 (32.1) | 340 (32.1) |
| ≥80 | 382 (26.0) | 3819 (26.0) | 90 (25.7) | 899 (25.7) | 58 (21.8) | 580 (21.8) | 34 (32.1) | 340 (32.1) |
| Median | 73.4 | 73.4 | 73.0 | 73.0 | 73.0 | 73.0 | 75.3 | 75.3 |
| **Survival 5 years (95% CI)†** | | | | | | | | |
| Overall | 72.3 (69.4 to 74.9) | 79.5 (78.7 to 80.3) | 69.6 (63.3 to 75.0) | 81.3 (79.7 to 82.9) | 74.2 (67.3 to 79.8) | 82.4 (80.6 to 84.1) | 68.6 (56.5 to 78.0) | 77.6 (74.1 to 80.7) |
| Relative | 90.6 (86.9 to 93.2) | 99.2 (98.3 to 99.7) | 84.2 (75.8 to 89.8) | 99.3 (93.7 to 99.9) | 88.3 (78.5 to 93.8) | 98.5 (95.3 to 99.5) | 77.6 (62.6 to 87.2) | 96.8 (89.5 to 99.1) |
| **Progressions/transformations†** | | | | | | | | |
| Total | 44 (3.0) | – | 8 (2.3) | – | 17 (6.4) | – | 6 (5.7) | – |
| Myeloma | 34 (2.3) | – | 1 (0.3) | – | 11 (4.1) | – | 2 (1.9) | – |
| Other | 10 (0.7) | – | 7 (2.0) | – | 6 (2.3) | – | 4 (3.8) | – |
| **Non-haematological cancer registrations†** | | | | | | | | |
| Total | 118 (8.0) | 1173 (8.0) | 41 (11.7) | 277 (7.9) | 19 (7.1) | 210 (7.9) | 7 (6.6) | 88 (8.4) |
| Lung | 16 (1.1) | 133 (0.9) | 6 (1.7) | 24 (0.7) | 4 (1.5) | 21 (0.8) | 5 (4.7) | 12 (1.1) |
| Prostate‡ | 14 (1.9) | 115 (1.5) | 3 (1.7) | 27 (1.5) | 3 (2.1) | 25 (1.7) | – | 11 (1.7) |
| Colorectal | 16 (1.1) | 112 (0.8) | 2 (0.6) | 26 (0.7) | 1 (0.4) | 20 (0.7) | – | 8 (0.8) |
| Breast‡ | 9 (1.2) | 88 (1.2) | 4 (2.3) | 26 (1.5) | 2 (1.7) | 15 (1.2) | – | 6 (1.4) |
| **Hospital outpatient activity (≥3 visits), excluding haematology and visits within 1 month of diagnosis§** | | | | | | | | |
| 5 years before diagnosis | 1173 (79.7) | 9103 (61.9) | 287 (82.0) | 2155 (61.6) | 216 (81.2) | 1633 (61.4) | 89 (84.0) | 663 (62.5) |
| 3 years after diagnosis | 964 (65.5) | 6558 (44.6) | 240 (68.6) | 1577 (45.1) | 176 (66.2) | 1154 (43.4) | 70 (66.0) | 445 (42.0) |

*Biclonal, IgD, IgE, light chain, not known.
†Followed up until 15 March 2017.
‡Prostate for men only, breast for women only.
§In the 5 years before diagnosis/pseudodiagnosis and 3 years after diagnosis/pseudodiagnosis.
HMRN, Haematological Malignancy Research Network.

and respiratory medicine. By contrast, although hospital activity increased in the months around the time of MBL diagnosis, no consistent differences or patterns either before or after the diagnosis were detected. Furthermore, in agreement with findings reported in other studies that used age-matched and sex-matched controls, no associations with mortality were detected.[3 32] However, given the fact that MBL has been associated with increased susceptibility to infection and non-CLL related mortality,[4 15 33] it is possible that the findings relating to subsequent morbidity and mortality could change as our data mature, length of follow-up increases and linkage to primary care data becomes possible.

The age and sex distributions of our population-based cohorts are broadly similar to those of other published MBL[3 34 35] and MGUS[1 2 11] series, as is the dominance of the IgG MGUS subtype.[1 2] Providing nationally generalisable data, additional strengths of our study include its large well-defined population, within which all haematological malignancies and related clonal disorders are diagnosed, monitored and coded using up-to-date standardised procedures at a central haematopathology laboratory.[19] In this context, it is important to bear in mind that most people with premalignant clonal disorders remain asymptomatic and that our cohorts contain a relatively large proportion of people who came to clinical attention in primary and/or secondary care and were referred to haematology for further investigation. More specifically, around 98% of patients in our MBL cohort had high-count MBL, and within the HMRN region, patients with MBL are monitored routinely using flow cytometry so laboratory progressions (B-cell count $>5\times10^9$/L) may be detected with higher sensitivity than in cases monitored clinically. This is supported by the fact that over the follow-up period (median 3.8 years), around a quarter of patients with MBL progressed to CLL and 3.6% required treatment. Prevalence comparisons with population screening studies also confirm that the majority of those over 50 years of age with monoclonal immunoglobulin in their blood/urine would not be included in our MGUS cohort.[5 20 35–37] In this context, it is important to remember that some members of the control cohort would, if screened, have had a premalignant clonal disorder detected in their peripheral blood.[13 32 33 38 39] Unfortunately, however, information on diagnostic work-up tests and monitoring procedures is not routinely included in nationally compiled HES. Furthermore, the anonymised nature of the control cohort means that individuals cannot be linked to other data sources. Hence, although we know that members of the control cohort had no prior record of an MGUS or MBL diagnosis within the study region in the years leading up to their corresponding case's diagnosis, we do not know how many people developed these conditions after their case was diagnosed.

The diversity of morbidity effects seen among individuals with MGUS is consistent with the expanding body of evidence relating to the potential adverse impact that even low levels of circulating monoclonal protein (M-protein) can have. Thus far, the complex underpinning mechanisms identified include: deposition of M-protein aggregations of varying immunoglobulin subtypes in different organs as well as the induction of autoantibodies and cytokines that can have an impact on organs and tissues in a variety of deleterious ways.[13 18 40 41] Indeed, the recognised number of M-proteinmediated entities is increasing, with several affecting multiple organs; well-known deposition syndromes including primary amyloidosis and paraneoplastic conditions such as POEMS syndrome.[7 42] As evidenced in our analysis, kidney involvement is frequent, both in the years before (fourfold excess) and after (14-fold excess) MGUS diagnosis. Indeed, the umbrella term monoclonal gammopathy of renal significance has recently been suggested to cover all M-protein-mediated kidney disorders that fail to meet the diagnostic criteria for multiple myeloma or any other B-cell malignancy.[13 18 43] Other organ-specific terms continue to emerge, and with a view to improving recognition of these complex disorders which clearly pose significant diagnostic and treatment challenges, the overarching term monoclonal gammopathy of clinical significance has also been suggested.[18]

From a haematological malignancy perspective, MGUS and MBL are generally considered to be relatively benign conditions. However, both can have other deleterious health consequences, the effect of monoclonal gammopathy being particularly striking. Impacting significantly the survival and having the potential to cause systemic disease and wide-ranging damage to most organs and tissues, the adverse outcomes associated with the M-proteins produced by the abnormal B-cell clone can be severe and extend over many years. Even though most people with monoclonal immunoglobulins never develop a B-cell malignancy or suffer from any other form of M-protein-related organ/tissue-related disorder, the consequences for those that do can be extremely serious. In this regard, early targeting of pathogenic B-cell clones could mitigate both cancer and non-cancer effects, but currently, although knowledge is increasing, there is no known way to reliably identify such clones in the absence of other signs/symptoms. Hence, population screening cannot be recommended and diagnosis remains reliant on clinical suspicion. However, the long-standing nature of the comorbidity associations seen prior to MGUS diagnosis in our data suggest that there may be room for improvement and that the implementation of strategies to improve awareness and earlier detection, as well as monitoring of high-risk patient groups, could prove beneficial.

**Author affiliations**
[1]Department of Health Sciences, University of York, York, UK
[2]Epidemiology and Prevention Programme, Uganda Virus Research Institute, Entebbe, Uganda
[3]Haematology, Leeds Teaching Hospitals NHS Trust, Leeds, UK
[4]Haematological Malignancy Diagnostic Service (HMDS), Leeds Teaching Hospitals NHS Trust, Leeds, UK
[5]Haematology, Hull University Teaching Hospitals NHS Trust, Hull, UK

**Contributors** ER, AS, DH and RP initiated the patient cohort within which this research is nested. ER, AS and EK designed the comparison cohort. ER, AS and MJEL planned this study and analyses. RdT and AR oversaw all laboratory procedures. MJEL, DP, EK and TB managed the data and carried out data analysis. GC, RP, AR and RN commented on the biological and clinical aspects. ER and MJEL drafted the manuscript, which was approved by all authors.

**Funding** This work was supported by Cancer Research UK, grant numbers 18362 and 29685, and Blood Cancer UK, grant number 15037.

**Competing interests** None declared.

**Patient and public involvement** Patients and/or the public were involved in the design, or conduct, or reporting, or dissemination plans of this research. Refer to the Methods section for further details.

**Patient consent for publication** Not required.

**Ethics approval** The Haematological Malignancy Research Network has ethics approval (REC 04/01205/69) from Leeds West Ethics Committee, R&D approval from each NHS Trust and exemption from Section 251 of the Health & Social Care Act (PIAG 1-05 9h)/2007).

**Provenance and peer review** Not commissioned; externally peer reviewed.

**Data availability statement** No data are available. Ethical approvals and data restrictions mean that data cannot be shared, but collaborative projects can be undertaken. The corresponding author can be contacted for more information.

**ORCID iDs**
Maxine JE Lamb http://orcid.org/0000-0002-1284-9912
Alexandra Smith http://orcid.org/0000-0002-1111-966X
Daniel Painter http://orcid.org/0000-0002-3936-7569
Eleanor Kane http://orcid.org/0000-0002-7438-9982
Timothy Bagguley http://orcid.org/0000-0002-6150-3467
Robert Newton http://orcid.org/0000-0001-6715-9153
Debra Howell http://orcid.org/0000-0002-7521-7402
Eve Roman http://orcid.org/0000-0001-7603-3704

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
