## [Reviewer comments · BMJ Open]

ARTICLE DETAILS

TITLE (PROVISIONAL)	Health impact of monoclonal gammopathy of undetermined significance (MGUS) and monoclonal B-cell lymphocytosis (MBL): findings from a UK population-based cohort
AUTHORS	Lamb, Maxine; Smith, Alexandra; Painter, Daniel; Kane, Eleanor; Bagguley, Timothy; Newton, Robert; Howell, Debra; Cook, Gordon; de Tute, Ruth; Rawstron, Andrew; Patmore, Russell; Roman, Eve

VERSION 1 – REVIEW

REVIEWER	Alberto Orfao University of Salamanca Spain
REVIEW RETURNED	23-Jul-2020

GENERAL COMMENTS	In this paper, Lamb et al investigate the health impact in terms of both morbidity and mortality before and after diagnosis of MGUS and MBL. The study relies on a large series of around 3000 patients, each matched with 10 subjects from a UK-population based cohort. While MGUS was associated with increased morbidity and mortality rates together with a low rate of progression to malignant disease, MBL showed a higher rate of progression to CLL in the absence of a significantly increased morbidity and mortality rates. The manuscript presents original information of great clinical value, with important impact on population health. Specific Comments: 1.- Authors should more clearly specify the selection criteria for the individuals in the population-based (control) cohort, and specify whether they had undergone tests to exclude MGUS (e.g. plasma protein electrophoresis) or MBL (Flow cytometry and/or blood count screening). If not screened authors should discuss the lack of differences on the light of the relatively lower vs higher frequency of MGUS vs MBL in the general population.2.- For the diagnosis of MGUS (vs its malignant counterparts) a BM study, as well as additional tests in case of amyloidosis, is required. In reading the methods section it appears that diagnoses were definitive; however, in one third of patients, no BM had been performed. The criteria and tests applied to e.g. rule out amyloidosis should be more clearly specified. Alternatively, the potential limitations of the inclusion criteria used and their impact on morbidity and mortality should be more clearly discussed.3.- From the paper one would understand controls per MGUS and MBL subject were selected according to their status at the time of patient diagnoses. In such case, the outcome of controls in terms of progression of disease should be provided. If this is not the case, and controls were selected at time of closing the 3y follow-up, then this should be clearly specified.
--

	4.- Authors should discuss and provide potential explanations for the very high rate (25% vs expected 3-6%) of progression observed from MBL to CLL. 5.- Regarding progression, the paper would benefit from an analysis of the frequency of patients requiring therapy for their neoplastic B cell condition, as transformation from MGUS usually includes progression to symptomatic disease with need for therapy, while progression of MBL to CLL is known to be mostly a biological progression (lymphocyte counts below vs above an arbitrary cut-off of 5000 cells/microliter) without symptoms and no therapeutic intervention that would potentially increase also morbidity. 6.- Please comment on the peak of activity around (prior and after) the date of diagnosis.
--	--

REVIEWER	Mariana Ciocchini Academia Nacional de Medicina de Buenos Aires
REVIEW RETURNED	21-Aug-2020

GENERAL COMMENTS	I've found your paper very interesting. These are the things that you must correct. Best regards. Page 3: Abstract Line 5: Mortality should be added to mortality. Line 8: Activity should be replaced by complications all along the paper. Line 22: Survival should be replaced by mortality. Line 37. The abstract conclusion should be rewritten because it's not very clear. Article summary This section should be deleted The strength and limitations of the study must be analysed in the discussion section. Background It should have more information. Methods Page 6, line 19: The definitions of the diseases are not complete. This is a very important subject. Discussion It should be rewritten. The last two paragraphs belongs to the introduction Page 11: Delete from line 34 to 42. Page 12, Line 4 :Make clear why not all the patients have a bone marrow biopsy. It wasn't necessary, .It wasn't possible....
---

VERSION 1 – AUTHOR RESPONSE

Reviewer: 1

Reviewer Name: Alberto Orfao

Institution and Country: University of Salamanca, Spain

Please state any competing interests or state 'None declared': None declared

In this paper, Lamb et al investigate the health impact in terms of both morbidity and mortality before

and after diagnosis of MGUS and MBL. The study relies on a large series of around 3000 patients, each matched with 10 subjects from a UK-population based cohort. While MGUS was associated with increased morbidity and mortality rates together with a low rate of progression to malignant disease, MBL showed a higher rate of progression to CLL in the absence of a significantly increased morbidity and mortality rates.

The manuscript presents original information of great clinical value, with important impact on population health.

Response: We thanks the reviewer for this comment

Specific Comments:

1.- Authors should more clearly specify the selection criteria for the individuals in the population-based (control) cohort, and specify whether they had undergone tests to exclude MGUS (e.g. plasma protein electrophoresis) or MBL (Flow cytometry and/or blood count screening). If not screened authors should discuss the lack of differences on the light of the relatively lower vs higher frequency of MGUS vs MBL in the general population.

Response: We thank the reviewer for pointing out this omission, and agree that more detailed information about the population-based cohort is warranted. Briefly, controls were randomly selected for all 18127 HMRN's patients diagnosed with a haematological malignancy or related disorder 2009-15. The selection (from the national population-based NHS Central Register) and linkage to national administrative databases (HES – Hospital Episode Statistics, cancers, and deaths) was done by NHS Digital (the national information and technology partner of the health and care system). The control dataset we hold does not contain personal identifiers (see response to 3 below), and HES does not contain information on diagnostic workups. We have clarified this in the text, and the second paragraph of the Methods section now reads as follows:

“To facilitate comparisons with the unaffected individuals, HMRN also has a general population cohort. To create this ‘control’ cohort, all patients diagnosed 2009-15 with a haematological malignancy or related clonal disorder (n=18,127) were individually matched on sex and age at the point of diagnosis to 10 randomly selected individuals from the same catchment population. All controls were assigned a serial number that linked them to their matched case, and a “pseudo-diagnosis” date that corresponded to their matched case’s diagnosis date. Individuals in the patient cohort and the comparison cohort are linked to the same nationwide information on deaths, cancer registrations and Hospital Episode Statistics (HES). At the point of selection and matching, all of the controls were resident in the HMRN region and none of had a previous cancer registration for a haematological malignancy. [21,22]. Hence, in contrast to the patient cohort, no additional health information outside that contained within national administrative datasets was available for the controls.”

With respect to the differences between individuals in the control cohort and those who were diagnosed with premalignant clonal disorders (patient cohort), we have now added the following to the Discussion:

“.....Furthermore, given the asymptomatic nature of MBL and MGUS, some members of the control cohort would, if screened, be found to have a premalignancy detected.”

2.- For the diagnosis of MGUS (vs its malignant counterparts) a BM study, as well as additional tests in case of amyloidosis, is required. In reading the methods section it appears that diagnoses were definitive; however, in one third of patients, no BM had been performed. The criteria and tests applied to e.g. rule out amyloidosis should be more clearly specified. Alternatively, the potential limitations of the inclusion criteria used and their impact on morbidity and mortality should be more clearly discussed.

Response: Again, we thank the reviewer for drawing attention for the need for more clarity here. All individuals in the present study had been referred to haematology/hematopathology for further work-up, and MGUS was diagnosed using standard guidelines that are commonly applied in routine diagnostic settings around the world. Accordingly, following clinical screening and serum confirmation of MGUS, invasive bone marrow examinations are then carried out on patients where it is deemed necessary to test for evidence of plasma cell neoplasms or other lymphoproliferative disorders – this includes monoclonal immunoglobulin deposition diseases like amyloidosis and POEMS syndrome. The first paragraph of the Methods section has been changes as follow:

“Cases are from the Haematological Malignancy Research Network (HMRN; www.hmrn.org), a specialist UK registry established in 2004 to provide robust generalizable data to inform contemporary clinical practice and research across the country as a whole.[19,20] HMRN operates under a legal basis that permits data to be collected directly from health records without explicit consent. Set within a catchment population of around four million that is served by 14 hospitals and has a socioeconomic profile which is broadly representative of the UK as a whole, all haematological cancers and related conditions are diagnosed and coded by clinical specialists at a single integrated haematopathology laboratory, the Haematological Malignancy Diagnostic Service (www.hmds.info), using standardized diagnostic criteria and the latest WHO ICD-O-3 classification.[7] Specifically, in relation to the present report, which cover diagnoses made in 2009-15, MBL was defined by a peripheral blood monoclonal B-cell count $<5 \times 10^9/L$ in individuals that had no other features of a B-cell lymphoproliferative disorder; and MGUS by a serum paraprotein less than 30 g/L, and in those where a bone marrow examination was considered necessary following clinical examination, a clonal bone marrow plasma cells/lymphoplasmacytic infiltration of less than 10%. Hence, within the HMRN region region, as in other diagnostic settings [21,22], invasive bone marrow examinations in MGUS patients are generally only carried in when laboratory or clinical features are indicative of an underlying plasma cell neoplasm, lymphoproliferative disorder, monoclonal immunoglobulin deposition disease (e.g. amyloidosis), or conditions like POEMS syndrome. [7]”

The following has also been added to the end of the Results

“Finally, during the study period (2009-15), in addition to the detection of a paraprotein in peripheral blood, around 70% (1527/2193) of MGUS patients in our cohort had a confirmatory bone marrow examination taken to exclude an underlying neoplasm. The patient characteristics and secondary care activity patterns of those who had bone marrow examinations were, however, broadly similar to those who did not (data not shown)”

3.- From the paper one would understand controls per MGUS and MBL subject were selected according to their status at the time of patient diagnoses. In such case, the outcome of controls in terms of progression of disease should be provided. If this is not the case, and controls were selected at time of closing the 3y follow-up, then this should be clearly specified.

Response: The reviewer is correct (see answer to 1 above) the general population control cohort was selected from the national population-based NHS Central Register (which holds data on all individuals registered for health care with the NHS). All controls were assigned a “pseudo-diagnosis” date that corresponded to their matched case’s diagnosis date; and individuals in the patient cohort and the comparison cohort are linked to the same nationwide information on deaths, cancer registrations and Hospital Episode Statistics (HES). Unfortunately, no information on diagnostic work-up tests or other detailed clinical information is available HES – either before or after the date that controls were matched to their corresponding case. Furthermore, for ethical and GDPR reasons, individual identifiers (NHS numbers etc.) are held for periodic updating purposes by NHS Digital alone – we do not hold identifiers and cannot link to additional sources. As detailed in the response to question 1, this has now been clarified in the Methods.

4.- Authors should discuss and provide potential explanations for the very high rate (25% vs expected 3-6%) of progression observed from MBL to CLL.

Response: As detailed in the report, MBL patients within this study were not ascertained through a diagnostic pathway, not through population-wide screening i.e. all individuals had been referred to haematology/hematopathology for investigation. Hence, the MBL patients in our report are likely to differ in important respects from those in a screening dataset. Hence, although the level of progression is comparatively high, it is broadly consistent with that seen in other reports. We have reordered and added additional text and references to the second and third paragraphs of the Discussion; the second part of the third paragraph now reads:

“In this context, it is important to bear in mind that most people with premalignant clonal disorders remain asymptomatic, and that our cohorts contain a relatively large proportion of people who came to clinical attention in primary and/or secondary care and were referred to haematology for further investigation. More specifically, around 98% of patients in our MBL cohort had high-count MBL, and

within HMRN patients are monitored routinely using flow cytometry so laboratory progressions (B-cell count $>5 \times 10^9/L$) may be detected with higher sensitivity than in cases monitored clinically. This is supported by the fact that over the follow-up period (median 3.8 years), around a quarter of MBL patients progressed to CLL, and 3.6% required treatment. Prevalence comparisons with population screening studies also confirm that the majority of those over 50 years of age with monoclonal immunoglobulin in their blood/urine would not be included in our MGUS cohort.[5,20,35–37] Furthermore, in the context of population screening, it is also important to remember that some members of the control cohort would, if screened, be found to have premalignant clonal disorders detected in their peripheral blood . [13,32,33,38,39]”

The Response to Question 5 below is also of relevance here.

5.- Regarding progression, the paper would benefit from an analysis of the frequency of patients requiring therapy for their neoplastic B cell condition, as transformation from MGUS usually includes progression to symptomatic disease with need for therapy, while progression of MBL to CLL is known to be mostly a biological progression (lymphocyte counts below vs above an arbitrary cut-off of 5000 cells/microliter) without symptoms and no therapeutic intervention that would potentially increase also morbidity.

Response: We agree with the reviewer; and a detailed examination based on all patients diagnosed with premalignant clonal disorders over the 14 years 2004-2018 is currently ongoing. These analyses include more clinical data than is appropriate in the present report; where the main focus is the general population comparison (which by necessity is restricted to the 6-years 2009-15 for which we currently hold data on controls).

6.- Please comment on the peak of activity around (prior and after) the date of diagnosis.

Response: As requested, we have added;

1) the following text to the Results:

“In the period before diagnosis, patients with MGUS (Figure 1A) had consistently higher outpatient activity rates than their controls, the disparity increasing markedly during the months leading up to the formal diagnosis of MGUS by haematopathology where it remained high for about 12 months, before gradually falling and levelling out at higher level than before diagnosis.”

2) the following text to the second paragraph of the Discussion:

“...Corresponding to the period of diagnostic work-up, our data also demonstrate the pronounced increase in hospital activity in the months surrounding MGUS diagnosis; the highest activity being observed in the six months before and six months after diagnosis...”

Reviewer: 2

Reviewer Name: Mariana Ciocchini

Institution and Country: Academia Nacional de Medicina de Buenos Aires

Please state any competing interests or state 'None declared': None declared

Please leave your comments for the authors below

Dear authors,

I´ve found your paper very interesting. These are the things that you must correct.

Best regards.

Response: We thanks the reviewer for this comment